# Assessment of *Lippia origanoides* Essential Oils in a *Salmonella typhimurium, Eimeria maxima,* and *Clostridium perfringens* Challenge Model to Induce Necrotic Enteritis in Broiler Chickens

**DOI:** 10.3390/ani11041111

**Published:** 2021-04-13

**Authors:** Makenly E. Coles, Aaron J. Forga, Roberto Señas-Cuesta, Brittany D. Graham, Callie M. Selby, Álvaro J. Uribe, Blanca C. Martínez, Jaime A. Angel-Isaza, Christine N. Vuong, Xochitl Hernandez-Velasco, Billy M. Hargis, Guillermo Tellez-Isaias

**Affiliations:** 1Department of Poultry Science, University of Arkansas, Fayetteville, AR 72701, USA; mecoles@uark.edu (M.E.C.); ajforga@uark.edu (A.J.F.); rsenascu@uark.edu (R.S.-C.); bmahaffe@email.uark.edu (B.D.G.); mccreer@uark.edu (C.M.S.); vuong@uark.edu (C.N.V.); bhargis@uark.edu (B.M.H.); 2Promitec S.A., Bucaramanga, Santander 680001, Colombia; gerencia@promitec.com.co (Á.J.U.); innovacion@promitec.com.co (B.C.M.); nutricionanimal@promitec.com.co (J.A.A.-I.); 3Departamento de Medicina y Zootecnia de Aves, FMVZ, Universidad Nacional Autonoma de Mexico, Mexico City 4510, Mexico; xochitl_h@yahoo.com

**Keywords:** broiler chickens, essential oils, intestinal permeability, necrotic enteritis, performance

## Abstract

**Simple Summary:**

In recent years, a considerable expansion in the use of essential oils as feed additives in animal nutrition has occurred, mainly as an alternative to antibiotic growth promoters worldwide. The objective of the present research was to evaluate dietary supplementation of essential oils from *Lippia origanoides* (LEO) on the in vitro proliferation of *Clostridium perfringens* (CP)*,* as well as the performance, intestinal integrity, and necrotic enteritis lesions using a previously established necrotic enteritis challenge model in broiler chickens. The supernatant from digested feed supplemented with LEO and inoculated with CP reduced CP in vitro compared with the positive control. LEO showed a significant reduction of the harmful effects of induced infection/dysbiosis and a significant reduction in NE lesion scores, morbidity and mortality compared with the positive challenge control group These results suggest that the dietary inclusion of *L. origanoides* could mitigate some of the complex negative impacts caused by necrotic enteritis.

**Abstract:**

The objective of the present research was to evaluate dietary supplementation of essential oils from *Lippia origanoides* (LEO) on necrotic enteritis (NE). Chickens were randomly assigned to three groups. Group 1: negative control; Group 2: positive control challenged with *Salmonella typhimurium* (day 1), *Eimeria maxima* (day 18), and *C. perfringens* (CP, days 22-23); Group 3: dietary supplementation LEO and challenged. On d 25 of age, serum samples were collected to evaluate fluorescein isothiocyanate-dextran (FITC-d), superoxide dismutase (SOD), gamma interferon (IFN-γ), Immunoglobulin A (IgA). Group 3 showed a significant reduction of the harmful effects of induced infection/dysbiosis and a significant reduction in NE lesion scores, morbidity and mortality compared with the positive challenge control group (*p* < 0.05) compared with Group 2. Digested feed supernatant, supplemented with LEO and inoculated with CP, reduced CP burden (*p* < 0.05). Group 3 also exhibited a significant reduction in FITC-d, IFN-γ and IgA compared with Group 2. However, a significant increase SOD was observed in Group 3 compared with both control groups. Further investigation to compare the effect of LEO and the standard treatment of clostridial NE is required.

## 1. Introduction

In recent years, a considerable attention to essential oils (EO) as nutraceuticals in livestock production has occurred, mainly as an alternative to antibiotic growth promoters (AGPs) worldwide. Essential oils are derived from various plants as secondary metabolites with well-documented antibacterial [1], antiviral [2], antifungal [3], antioxidant [4], digestive stimulants [5], and immunomodulatory properties [6]. Some EO are used in a combination with other phytochemicals to increase performance in poultry [3]. Hence, EO have played an important role in controlling the increased incidence of coccidiosis and necrotic enteritis (NE) following the removal of ionophores and AGPs [7]. Since NE is a multifactorial disease involving *Eimeria* spp. and *Clostridium perfringens*, in many cases, EO are combined with other strategic products such as probiotics, prebiotics, organic acids, and enzymes to modulate the intestinal microbiota and immune system of the birds [7,8,9].

In poultry, NE is a disease caused by the anaerobe Gram-positive bacterium, *C. perfringens* (CP). Several virulence factors have been identified in the pathogenesis of NE, including toxins, proteolytic enzymes, bacteriocins, and adherent molecules produced by virulent strains of CP [10]. In 2000, the annual worldwide economic cost of NE was estimated at over USD 2 billion [11,12]. However, the global number of chickens has increased from 14.38 billion chickens in 2000 to 23.70 billion chickens in 2018 [13]. Based on those statistics, annual losses associated with mortality and decreases in performance caused by NE at the end of 2020 can be estimated at over USD 4 billion, especially in subclinical cases of NE [14,15]. Several factors have been identified in the pathogenesis of this multifactorial disease such as *Eimeria* spp. infections, immunosuppression, dysbacteriosis, and removal of AGPs [7,8,9,10]. Hence, several investigators have evaluated the use of several nutraceuticals that include EO [16], organic acids [17], probiotics [18], and plant extracts [19] as an effort to reduce the economic impact that NE has in the modern poultry industry.

A recent study has shown that early microbiota disruption (dysbiosis) is a major consequence of necrotic enteritis [20]. In neonate chickens, ST is a primary pathogen responsible for severe intestinal pathology associated with inflammation, dysbiosis, tight junction (TJ) disruption, and immune suppression [21,22,23]. Our early studies confirmed that those conditions following a challenge with *Eimeria maxima* (EM) and CP induce a reliable laboratory model for NE. The objective of the present research was to evaluate dietary supplementation of *Lippia origanoides* essential oils (LEO) on in vitro proliferation of *C. perfringens*, as well as performance, intestinal integrity, and NE lesions using early *Salmonella typhimurium* (ST) infection as a predisposing factor for NE in broiler chickens [18,24,25,26].

## 2. Materials and Methods

### 2.1. Ehics

All animal handling procedures complied with the Institutional Animal Care and Use Committee (IACUC) at the University of Arkansas, Fayetteville. Explicitly, the IACUC approved this study under protocol # 21018.

### 2.2. Essential Oils from Lippia origanoides

The LEO (Natbio EsencialPremix^®^) was provided by Promitec S.A. (Bucaramanga, Santander, Colombia) and feed inclusion based on the manufacturer’s recommendations. The product contains EO of *Lippia origanoides* microencapsulated with maltodextrin by spray drying with a diet inclusion rate of 37 ppm [27,28]. The qualitative and quantitative chemical composition of *Lippia origanoides* essential oils of LEO is summarized in Table 1. The LEO sample was submitted to chromatographic analysis 7890A (Laboratory of chromatography and mass spectrometry Industrial University of Santander, Bucaramanga, Colombia). The sample showed 16 compounds, however, Carvacol and thymol were the compounds with the highest content (Table 1). The LEO was included in all three diets and administered since day 1. Starter, grower, and finisher mash diets were used in this experiment and were formulated to approximate the nutritional requirements of broiler chickens as recommended by the National Research Council [29] and adjusted to the breeder’s recommendations [30]. No antibiotics, coccidiostats or enzymes were added to the feed (Table 2). 

The LEO (Natbio EsencialPremix^®^) sample was submitted to chromatographic analysis 7890A (Laboratory of chromatography and mass spectrometry Industrial University of Santander, Bucaramanga, Colombia)

### 2.3. Animal Source and Experimental Design

In the present study, an experiment was conducted to evaluate the effect of *L. origanoides* LEO on a NE model previously described by our laboratory [25], which was confirmed and extended in more recent studies [18,24,26]. Three hundred one-day-old male Cobb-Vantress broiler chickens were obtained from a commercial hatchway (Fayetteville, AR, USA). Upon arrival, all chickens were neck tagged, weighed and randomly allocated to one of three groups with ten replicates (n = 10 chickens/replicate). Chickens were placed in battery cages, with a controlled age-appropriate environment. Group 1: negative control (NC); Group 2: positive control (PC) challenged with ST (day 1), EM (day 18), and CP (days 22 and 23); and Group 3: 37 ppm final feed concentration LEO, challenged in the same manner as the PC. Chicks received ad libitum access to water and feed for 25 days. Chickens received 23 h of light from days 1 to 4, 20 h of light from days 5 to 14, and 18 h of light from days 15 to 25. Light intensity was set at 30-footcandle the first week, 1-foot candle from days eight to fourteen, and 0.5-footcandle from days 15 to 25. Body weight (BW) was recorded on days 0, 7, 14, 18, and 25. Body weight gain (BWG) was recorded on d 0–7, 8–25, and 0–25. Feed intake (FI) and feed conversion ratio (FCR) were recorded on d 0–7, 8–25, and 0–25. All performance parameters were calculated using a battery cage as a unit (n = 10 replicates).

### 2.4. Evaluation of Serum Levels of FITC-d, SOD, IFN-γ, and IgA

At the end of the trial, on d 25 of age, two random chickens per cage were selected (n = 20) and orally gavaged with 8.32 mg/kg of body weight of fluorescein isothiocyanate-dextran (FITC-d, MW 3–5 KDa; Sigma-Aldrich Co). One hour after FITC-d administration, chickens were euthanized by CO_2_ inhalation. Blood samples were collected from the femoral vein and centrifuged (1000× *g* for 30 min at 4 °C) to separate the serum. Serum levels of FITC-d (ng/mL) were used as a biomarker to evaluate intestinal permeability as described by Baxter et al. [31]. Commercial kits were used to evaluate serum levels of superoxide dismutase (SOD, U/mL, (Cayman Chemical Company, Ann Arbor, Michigan, USA), gamma interferon (IFN-γ, pg/mL, Invitrogen Corporation, Frederick, Maryland, USA). Immunoglobulin A (IgA, ng/mL) as described by Merino-Guzman et al. [32]. 

### 2.5. Necrotic Enteritis Lesion Score

To evaluate Ileal NE lesion score, four chickens per each cage, separately from those in Section 2.3 (n = 40 chickens/group), were euthanized by CO_2_ inhalation. Lesion score was evaluated as previously described by Hofacre et al. [33] where 0 = no lesions; 1 = thin-walled and friable intestines; 2 = focal necrosis, gas production, and ulceration; 3 = extensive necrosis, hemorrhage, and gas-filled intestines; and 4 = generalized necrosis typical of field cases and marked hemorrhage. 

### 2.6. Necrotic Enteritis Model: Challenge Organisms

The challenge organisms ST, EM- GS, and CP isolate details and culture conditions used to induce NE are described in previous studies [18,24,25,26,33]. One-day-old broiler chickens were weighed and challenged with 1 × 10^8^ cfu of ST per bird by oral gavage. At day 18, chickens in challenged groups were orally gavage with 40,000 oocyst per mL. The dose was selected based on a previous trial conducted to determine a challenge dose causing sub-clinical coccidiosis and reduction of 35% body weight gain as described previously [18,24,25,26]. The *C. perfringens* culture was administered on days 22 and 23 of age via oral gavage at a concentration of 1 × 10^9^ cfu per bird and was also used in the in vitro proliferation assay (described below).

### 2.7. Clostridium perfringens Proliferation Using In Vitro Digestion Assay

The in vitro digestion model used in the present study was based on previous publications, with minor modifications [34,35]. The test was conducted using from the two experimental diets, with or without LEO (37 ppm) in quintuplicates (n = 5). Briefly, for all the gastrointestinal compartments simulated during the in vitro digestion model, a biochemical oxygen demand incubator (VWR, Houston, TX, USA) set at 40 °C (to simulate poultry body temperature), customized with a standard orbital shaker (19 rpm; VWR, Houston, TX, USA) was used for mixing the feed content. Additionally, all tube samples were held at an angle of 30° inclination to facilitate proper blending of feed particles and the enzyme solutions in the tube. The first gastrointestinal compartment simulated was the crop, where 5 g of feed and 10 mL of 0.03 M hydrochloric acid (HCL, EMD Millipore Corporation, Billerica, MA, USA) were placed in 50 mL polypropylene centrifuge tubes and mixed vigorously reaching a pH value around 5.2. Tubes were then incubated for 30 min. Following this time, all tubes were removed from the incubator. To simulate the proventriculus as the next gastrointestinal compartment, 3000 U of pepsin per g of feed (Sigma-Aldrich, St Louis, MO, USA) and 2.5 mL of 1.5 M HCl were added to each tube to reach a pH of 1.4 to 2.0. All tubes were incubated for additional 45 min. The third and final step was intended to simulate the intestinal section of the gastrointestinal tract. For that, 6.84 mg of 8 × pancreatin (Sigma-Aldrich, St Louis, MO, USA) in 6.5 mL of 1.0 M sodium bicarbonate (Sigma-Aldrich, St Louis, MO, USA) were added, and the pH was adjusted to range between 6.4 and 6.8 with 1.0 M sodium bicarbonate. All tube samples were further incubated for 2 h. Hence, the complete in vitro digestion process took 3 h and 15 min. After the digestion, supernatants from all the diets were obtained by centrifugation for 30 min at 2000 g. All supernatants were then tested for CP proliferation. Experimental groups were: (Group 1) 3 mL tryptic soy broth (TSB, Becton Dickinson, Sparks, MD, USA) supplemented with sodium thioglycolate (THIO, Sigma-Aldrich, St Louis, MO, USA) + 3 mL supernatant from digested control diet without CP inoculation (negative control); (Group 2) 3 mL TSB supplemented with THIO + 3 mL supernatant from digested control diet inoculated with 10^5^ cfu/mL of CP (positive control); or (Group 3) 3 mL TSB supplemented with THIO + 3 mL supernatant from a digested diet supplemented with LEO inoculated with 10^5^ cfu/mL of CP. Samples were incubated anaerobically at 40 °C, with tubes set at a 30° angle with continuous shaking (200 rpm) for four hours. Following incubation, ten-fold serial dilutions were made from all three experimental groups, plated on TSA supplemented with THIO and incubated for twenty-four hours at 40 °C, anaerobically. Results are reported as log_10_ cfu of CP per mL.

### 2.8. Data and Statistical Analysis

For performance variables, each group had ten replicates and ten chickens in each replicate (n = 10 replicates/group; 10 chickens/replicate). BW, BWG, FI and FCR were calculated by cage bases (n = 10). The number of samples per variable evaluated was n = 5 for in vitro proliferation of CP; n = 40 for NE lesion score; n = 20 for serum variables. Hence, all these variables implied a normal distribution (Shapiro–Wilk test), and the homoscedasticity was verified (Levene’s test). Accordingly, the parametric test of analysis of variance as a completely randomized design, using the General Linear Models procedure of SAS was used [36]. Significant differences among the means were determined by Duncan’s multiple range test at *p* < 0.05, and the *P* value was established with an alpha level of *p* < 0.05. Total mortality were compared by a chi-square test of independence, testing all possible combinations to determine the significance with an alpha level of *p* < 0.05.

## 3. Results

### 3.1. Effect of LEO on Growth Performance of Chicken Challenged with NE Organisms

The evaluation of body weight, body weight gain, feed intake, and feed conversion ratio in broiler chickens consuming a diet supplemented with 37 ppm LEO on a necrotic enteritis challenge model are summarized in Table 3. Both groups of chickens that were challenged with ST at d 1 showed a reduction (*p* < 0.05) in BW at d 7 when compared with negative control chickens. It was interesting to observe that chickens that received the diets with the inclusion of LEO had a similar BW compared with negative control chickens on d 14 and 18, before *E. maxima* challenge. However, BW of and BWG of positive control chickens that were challenged with EM and CP were severely affected (Table 3). At the end of the trial, chickens receiving LEO showed minimal but statistically improvement in BW and BWG when compared with the positive challenge control chickens (Table 2). As expected, negative control chickens showed an increase in FI and more efficient FCR as compared with chickens that were challenged. In this study, chickens that received LEO had an improvement in FI compared with positive control chickens but similar FCR (Table 3). Several studies have shown that dietary supplementation with EO improve performance parameters in other NE models [15,16,17,18,19]. 

### 3.2. Antimicrobial Effect of LEO on Clostridium perfringens log_10_ Count Using In Vitro Digestion Assay

Table 4 shows the evaluation of CP proliferation using in vitro digestion assay, NE lesion score, morbidity and mortality in broiler chickens consuming a diet supplemented LEO on a necrotic enteritis challenge model. The supernatant from digested feed supplemented with LEO and inoculated with CP showed a reduction in log_10_ cfu/mL of CP when compared with the positive control feed without LEO. Supernatant from digested feed without CP inoculation remain negative to CP (Table 4).

### 3.3. Protective Effect of LEO against NE Lesion in Chicken Challenged with NE Organisms

Chickens that consumed LEO-supplemented feed had a reduction in NE lesion scores compared with positive challenge control chickens. No lesions were observed in negative controls (Table 4).

### 3.4. Morbidity and Mortality

At d 18, before the NE challenge, all groups showed no clinical signs. However, at d 24, chickens that received the challenges of EM and CP exhibited clinical signs that included reduced activity and pronounced lethargy. Chickens in the positive control group had an increase mortality due to NE (8.33%) compared to 1.25% in chickens that received LEO. No mortality due to NE in the negative control group. Overall, chickens in the positive control group had also an increase in mortality rate compared with negative control chickens, or chickens that received LEO (Table 4).

### 3.5. Improvement of Intestinal Integrity, Antioxidant, and Anti-Inflammatory Effect of LEO in Chicken Challenged with NE Organisms

The results of the evaluation of serum levels of FITC-d, SOD, IFN-γ, and IgA in broiler chickens consuming a diet supplemented LEO on a NE challenge model are summarized in Table 5. A reduction in FITC-d serum levels was observed in chickens in the negative control group when compared with chickens that were challenged. Interestingly, chickens that received LEO showed a reduction in leakage of FITC-d from the intestine into the bloodstream when compared with positive challenge controls. Furthermore, an increase in serum levels of SOD was also observed in chickens that received LEO compared with both control groups. Interestingly, chickens receiving LEO showed a reduction in serum levels of IFN-γ and IgA compared with positive control chickens (Table 5). 

## 4. Discussion

Recent studies have demonstrated that the immune modulator properties of synbiotics significantly reduced ST infection, and such reduction was associated with reduction in the severity of NE caused by CP in broilers chickens challenged with both enteropathogens [37,38].

It is estimated that EO are produced by over 17,500 species of plants; however, only a minor fraction are used commercially [39]. The chemical composition of EO include several terpenes, terpenoids, and phenylpropanoids. Additionally, sulfur derivatives, fatty acids, aldehydes, alcohols, and oxides have also been identified [1,40].

Due to their antimicrobial properties, the use of whole EO is more advantageous than purified components due to synergistic effects, multiple biological properties, and fewer probabilities to select for antimicrobial resistance [1,4,39,41]. One particular study has emphasized the antimicrobial effects of essential oils, even against multi-drug resistant bacteria [42]. In another recent study, flow cytometry showed EO’s antimicrobial properties are due to the increase in the permeability of the bacterial membrane and inhibition of the efflux pump activity [6]. In the same study, the immunomodulatory effect of EO on THP-1 cells was also demonstrated by gene expression profiles of pro-and anti-inflammatory cytokines [6]. Of equal importance, EO biological properties have been associated with maintaining intestinal integrity [43], strengthening the mucosal barrier [44], and microbiota modulation [45].

Even at low concentrations, EO have shown substantial and superior antibacterial properties against *C. perfringens* and other pathogenic bacteria compared to antibiotics [46]. Similarly, in the present study, feed inclusion of 37 ppm LEO significantly reduced the proliferation of *C. perfringens* in the digested supernatant of an in vitro digestive model compared to the positive control diet by 1.94 logs. While EO have shown enhanced antimicrobial activity, the encapsulation of EO has shown higher antimicrobial activity compared to the use of EO alone by slowing down the degradation and reducing the organoleptic effects of EO [47,48]. In the present study, LEO was microencapsulated with maltodextrin through the spray dry process [27,28]. At an inclusion rate of 37 ppm, this formulation showed a significant reduction of the harmful effects of induced infection/dysbiosis and a significant reduction in NE lesion scores, morbidity and mortality compared with the positive challenge control group. These findings are in agreement with the results published by other studies [16,17]. Moreover, chickens that received LEO had a significant decrease in FITC-d serum levels compared to positive challenge control chickens. Due to its high molecular weight (3–5 KDa), intestinal leakage of FITC-d has been reported to be a reliable biomarker to evaluate intestinal inflammation and disruption of tight junction proteins in several poultry models published by our laboratory [49,50]. Other investigators showing a substantial improvement on intestinal integrity and permeability by oregano EO [51,52] have described similar results. Moreover, a recent review [53] describes the different mechanisms published by several researchers demonstrating that phytobiotics improve intestinal permeability by increasing the gene expression of tight junction (TJ) proteins, immunomodulation by augmenting gene expression of cytokines, and proliferation of goblet cells. These effects of pytobiotics used as feed additives in poultry diets suggest that productivity, health and welfare of poultry following ban or restriction of in-feed antibiotic use is possible. Depending upon the types of PBC, they possess antimicrobial, digestive enzyme secretion stimulation, antioxidant and many pharmacological properties, which are responsible for beneficial effects in poultry production. Moreover, they may also improve the intestinal barrier function and nutrient transport. 

Furthermore, there was a significant reduction in IFN-γ serum levels in chickens that received LEO compared with positive control chickens, suggesting a reduction in the inflammatory response [54,55]. Moreover, these serological results were also associated with a significant reduction in secretory IgA serum levels in chickens supplemented with LEO compared with the positive control (Table 4). Comparable results have been confirmed in previous studies conducted in our laboratory with the use of other phytobiotics and organic acids [56,57]. As indicated by Staley et al. [58], secretory IgA is an excellent biomarker to evaluate stress and animal welfare. However, studies conducted in our laboratory have shown that is a good biomarker to evaluate intestinal inflammation too [59].

In addition to their antimicrobial properties, other assets of EO are their antioxidant activity and radical scavenging activities [60]. In fact, the most studied assets of EO are their antioxidant activity, radical scavenging, and antimicrobial properties [60]. Essential oils from rosemary, oregano, thyme, and turmeric increased the antioxidant response element in enterocytes suffering oxidative stress, suggesting a unique mechanism by these compounds [61]. Oxidative damage may be minimized by antioxidant defense mechanisms that protect the cell against cellular oxidants and repair systems that prevent the accumulation of oxidatively damaged molecules. Antioxidant enzymes such as catalase (CAT), superoxide dismutase (SOD) and glutathione peroxidase (GPx) play a vital role in protecting cellular damage from harmful effects of ROS [62]. In the present study, chickens fed the formulation of microencapsulated LEO at 37 ppm had increased SOD serum concentrations compared with the positive control chickens (Table 4). Free radical scavenging capacity protects the integrity of cellular and mitochondrial membranes from lipid peroxidation [63,64]. The EO from *L. origanoides* have shown to increase the antioxidant response, radical scavenging activity, and apoptosis, suggesting a unique mechanism by these compounds [61,62,65,66,67,68].

## 5. Conclusions

At an inclusion rate of 37 ppm LEO, this formulation showed a significant reduction of in vitro CP proliferation as well as reduction of the harmful effects induced by the infection/dysbiosis of the NE model compared with the positive challenge control group. These effects were associated with the antimicrobial, anti-inflammatory, and antioxidant properties of LEO. However, further investigation to compare the effect of LEO and the standard antibiotic treatment of clostridial NE is required. It is fair to mention that this was a limitation in the present study. Moreover, it is imperative to establish standardized protocols that consider individual and inter-sample variability and consider the utility of molecular tools and epigenetic adjustments underlying phytobiotics such as EO, where research has not yet been elucidated.

## Figures and Tables

**Table 1 animals-11-01111-t001:** Qualitative and quantitative chemical composition of *Lippia origanoides* essential oils.

Compounds	Retention Time	*Lippia origanoides* Essential Oils (%)
α-Pinene	17.65	<0.1
β-Pinene	19.66	0.7
β-Myrcene	20.02	0.1
α-Phellandrene	21.95	0.1
ɣ-3-Carene	20.98	<0.1
α-Terpinene	21.31	0.2
p-Cymene	21.66	10.3
Limonene	21.85	0.2
β-Ocimene	22.46	<0.1
ɣ-Terpinene	23.08	10.5
β-Phellandrene	21.95	0.1
Terpinolene	24.20	<0.1
*p*-Cymenene	24.38	0.1
Diethylphenol	31.53	0.1
Thymol	31.70	47.5
Carvacrol	32.36	29.9

**Table 2 animals-11-01111-t002:** Ingredient composition and nutrient content of the corn-soybean diet used on an as-is basis.

Feed Ingredients	Stater Phase (d 1 to 7)	Grower Phase (d 8 to 14)	Finisher Phase (d 15 to 25)
Ingredients (%)			
Corn	51.80	57.81	59.64
Soybean meal (46.5% CP)	37.66	31.62	27.23
DDGS 8.1% EE	4.00	4.00	6.00
Poultry fat	3.24	3.44	4.38
Limestone	1.01	1.06	1.03
Dicalcium phosphate	1.00	0.88	0.64
Salt	0.35	0.35	0.31
DL-methionine	0.29	0.31	0.28
L-lysine HCl	0.12	0.13	0.12
Mineral premix ^a^	0.10	0.10	0.10
Vitamin premix ^b^	0.10	0.10	0.10
L-threonine	0.08	0.09	0.09
Choline chloride	0.06	0.06	0.05
Sodium bicarbonate	0.04	0.05	0.03
Antioxidant ^c^	0.15	0.15	0.15
Total	100	100	100
Calculated analysis			
ME (kcal/ kg)	3015.00	3090.00	3175.00
Ether extract (%)	5.88	6.20	7.28
Crude protein (%)	22.30	20.00	18.70
Lysine (%)	1.18	1.05	0.95
Methionine (%)	0.59	0.53	0.48
Threonine (%)	0.77	0.69	0.65
Tryptophan (%)	0.25	0.22	0.20
Total calcium (%)	0.90	0.84	0.76
Total phosphorus (%)	0.63	0.58	0.53
Available phosphorus (%)	0.45	0.42	0.38
Sodium (%)	0.20	0.20	0.18
Potassium (%)	1.06	0.94	0.87
Chloride (%)	0.27	0.28	0.25
Magnesium (%)	0.19	0.18	0.17
Copper (%)	19.20	18.46	18.85
Selenium (%)	0.28	0.27	0.26
Linoleic acid (%)	1.01	1.13	1.16

^a^ Vitamin premix supplied the following per kg: vitamin A, 20,000,000 IU; vitamin D3, 6,000,000 IU; vitamin E, 75,000 IU; vitamin K3, 9 g; thiamine, 3 g; riboflavin, 8 g; pantothenic acid, 18 g; niacin, 60 g; pyridoxine, 5 g; folic acid, 2 g; biotin, 0.2 g; cyanocobalamin, 16 mg; and ascorbic acid, 200 g (Nutra Blend LLC, Neosho, MO 64850). ^b^ Mineral premix supplied the following per kg: manganese, 120 g; zinc, 100 g; iron, 120 g; copper, 10–15 g; iodine, 0.7 g; selenium, 0.4 g; and cobalt, 0.2 g (Nutra Blend LLC, Neosho, MO 64850). ^c^ Ethoxyquin.

**Table 3 animals-11-01111-t003:** Evaluation of body weight (BW), body weight gain (BWG), feed intake (FI), and feed conversion ratio (FCR) in broiler chickens consuming a diet supplemented with 37-ppm *Lippia origanoides* essential oils on a necrotic enteritis challenge model *.

Item	Negative Control	Positive Control *	*Lippia origanoides* *	Pooled SEM	*p*-Value
BW, g/broiler					
d 0	42.44	41.55	42.12	6.10	0.1341
d 7	160.12 ^a^	137.44 ^b^	146.80 ^ab^	26.45	0.0002
d 14	455.10 ^a^	416.89 ^b^	440.03 ^ab^	50.33	0.0001
d 18	696.89 ^ab^	660.48 ^b^	687.12 ^ab^	31.81	0.0622
d 25	1181.16 ^a^	818.23 ^c^	851.03 ^b^	75.10	0.0001
BWG, g/broiler					
d 0–7	118.93 ^a^	97.46 ^b^	105.77 ^ab^	41.36	0.0002
d 8–25	1050.12 ^a^	653.45 ^b^	639.07 ^b^	99.24	0.0001
d 0–25	1143.09 ^a^	756.62 ^c^	810.44 ^b^	89.76	0.0002
FI, g/broiler					
d 0–7	142.91	132.93	131.41	12.65	0.1256
d 8–25	1180.75 ^a^	983.39 ^b^	901.94 ^c^	59.48	0.0001
d 0–25	1537.38 ^a^	1295.70 ^c^	1373.92 ^b^	116.45	0.0002
FCR					
d 0–7	0.91	0.94	0.89	0.88	0.1756
d 8–25	1.14 ^c^	1.36 ^b^	1.26 ^ab^	0.79	0.0002
d 0–25	1.36 ^c^	1.67 ^b^	1.75 ^ab^	0.81	0.0001

* Day-old broilers were challenged with *Salmonella typhimurium* (day 1), *E. maxima* (day 18), and *Clostridium perfringens* (days 22 and 23). ^a,b,c^ Non-matching superscripts within rows indicates significant difference at *p* < 0.05.

**Table 4 animals-11-01111-t004:** Evaluation of *Clostridium perfringens* proliferation using in vitro digestion assay, necrotic enteritis (NE) lesion score, morbidity and mortality in broiler chickens consuming a diet supplemented with 37-ppm *Lippia origanoides* essential oils on a necrotic enteritis challenge model *.

Item	Negative Control	Positive Control *	*Lippia origanoides* *	Pooled SEM	*p*-Value
*In vitro proliferation of C. perfringens* (log_10_ cfu/mL)	0.00 ^c^	6.95 ^a^	5.01 ^b^	0.14	0.0001
NE Lesion score	0.00 ^c^	2.63 ^a^	1.76 ^b^	0.58	0.0001
Morbidity day 18	0/86 (0%)	0/80 (0%)	0/81 (0%)		
Morbidity day 24	0/86 (0%) ^b^	78/78 (100%)	81/81 (100%)		
Mortality due to NE	0/86 (0%) ^b^	6/78 (8.33%) ^b^	1/81 (1.25%) ^a^		
Total mortality	14/100 (14%) ^b^	28/100 (28%) ^a^	20/100 (20%) ^b^		

* Day-old broilers were challenged with *Salmonella typhimurium* (day 1), *E. maxima* (day 18), and *Clostridium perfringens* (days 22 and 23). Morbidity and mortality data are expressed chickens with clinical signs of death/total number of chickens (%). ^a,b,c^ Non-matching superscripts within rows indicates significant difference at *p* < 0.05.

**Table 5 animals-11-01111-t005:** Evaluation serum levels of fluorescein isothiocyanate dextran (FITC-d), superoxide dismutase (SOD), gamma interferon (IFN-γ), and IgA in broiler chickens consuming a diet supplemented with 37-ppm *Lippia origanoides* essential oils on a necrotic enteritis challenge model *.

Item	Negative Control	Positive Control *	*Lippia origanoides* *	Pooled SEM	*p*-Value
FITC-D (ng/mL)	80.00 ^c^	540.00 ^a^	407.02 ^b^	49.32	0.0001
SOD (U/mL)	8.85 ^c^	10.28 ^b^	14.73 ^a^	8.56	0.0002
IFN-γ (pg/mL)	78.10 ^c^	281.4 ^a^	161.55 ^b^	24.23	0.0002
IgA (ng/mL)	11.95 ^b^	17.45 ^a^	10.45 ^c^	45.10	0.0001

* Day-old broilers were challenged with *Salmonella typhimurium* (day 1), *E. maxima* (day 18), and *Clostridium perfringens* (days 22 and 23). ^a,b,c^ Non-matching superscripts within rows indicates significant difference at *p* < 0.05.

## Data Availability

The datasets generated for this study are available on request to the corresponding author.

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
