# Peer review of "Assessment of Lippia origanoides Essential Oils in a Salmonella typhimurium, Eimeria maxima, and Clostridium perfringens Challenge Model to Induce Necrotic Enteritis in Broiler Chickens"

_animals, 2021, doi:10.3390/ani11041111_

Round 1
Reviewer 1 Report
Please supplement the pictures of intestinal lesions of broiler chickens NE positive control and the LEO group of broilers.
Author Response
Dear Reviewer, #1, thank you very much for the time you have spent on reviewing our manuscript. Your comments are very valuable and helpful for revising our paper and guiding our research. Sadly and unfortunately, in this study we did not take pictures of the intestinal lesions of the broiler chickens NE positive control and the LEO group. Indeed, a valuable lesson. In future studies, we will take pictures. However, considering the comments of Reviewers 2 and 3, we have made substantial changes in the manuscript. Thank you.
Reviewer 2 Report
- General comment: Please make sure to write in italics the Latin names of microorganisms in the whole manuscript and the references.
- General comment: The are too many references added in some parts of the manuscript that seem unnecessary. Line 73 has 11 references. Other examples with too many references are lines 53, 240, 252, 257, 279.
- Line 32: Correct the capital first letter of “Typhimurium” and use italics. Do the same in lines 75, 94, 115, 37
- Lines 74-86: The description how the Authors’ “laboratory” created this necrotic enteritis challenge model should be limited to the absolutely necessary, as this protocol is simply applied in this manuscript that focuses on Lippia origanoides.
- Table 1: Please revise the descriptions of the feed Ingredients (not “Items”). Write the full forms of SBM and DDGS. Delete numerical values i.e. “9-14-18”, “45.16%”, “8.1%”, “60%”, “0.06%”. Delete “0.5lb/ton”. Explain the composition of the various vitamin, mineral and other premixes (Waldroup TM mix? Tyson 2x Broiler Vit? Optiphos2000? Se Premix? Santoquin)
- Line 117. Place a dot after PC.
- Lines 128-130. Give details about the commercial kits (Names & Producer)
- Line 132: Text “n=40” refers to 4 chicken per cage? How were these slaughtered? Separately for those in section 2.3?
- Line 137: The abbreviations of these bacteria were already introduced previously in the manuscript. There is no need to repeat them here.
- Line 142: Correct “prvious”
- Lines 148-159: Give a brief description of the production of the “digested diets” and the inoculation procedure. Explain all uncommon abbreviations (THIO, TSB, TSA).
- Lines 166-177: Explain in more details the situation shown in Table 2. What happened before and after each inoculation?
However, the “beneficial” effect of essential oil supplementation although statistically significant is actually quite minimal. At day 25, Group 3 had only 36.67 g improved weight compared to positive control, but it had 323, 04 g lower weight that the negative control.
In addition, it would be very helpful to learn about the mortality of the birds. Were there no dead birds at all in groups 2 and 3 after the triple challenge?
- Table 2: It would be preferable to only write the superscripts for the parameters that differ significantly (P<0.05). Also, the addition of a new column with the P values of the ANOVA would provide additional info for the readers. Moreover, why did you not include the periods of days “8-14” and “15-25” (i.e. the feed change periods)? Or you could take into consideration periods up to & after day 18, since at day 18 the birds were inoculated with E. maxima.
Also, check values “4.8 2” and “12.0 5”, because the have a “space” between the last two digits.
- Figure 1: The quality of this figure is low. The text and numbers very small and hard to read. The “error lines” cannot be distinguished from the gray color of the bars. Either revise the figure or present these results in a table.
- Table 3: Did you take into consideration in your statistical analysis that the “negative control” has zero variation between its replications (±0.00)? Did you check the homogeneity of the data before applying the ANOVA? Probably a non-parametric test (such as Kruskal–Wallis test) should be applied to these kind of data instead of the ANOVA. Alternatively, a test excluding NC and only testing groups 2 (NC) and 3 (Lippia) should be performed.
Furthermore, this table combines results from two different procedures. The lesion scores are from the in vivo trial, whereas the C. perfringens are from the in vitro test. Perhaps they should be presented separately, with titles that clearly describe this point.
- Discussion section: Do not repeat things that were mentioned in the introduction (for example lines 219-221).
- Conclusions: Do not use references and comparisons with other works in the conclusions. Summarize and highlight the importance of your results.
Author Response
Dear Reviewer, #2, thank you very much for the time you have spent on reviewing our manuscript. Your comments are very valuable and helpful for revising our paper and guiding our research. We have studied those comments carefully and have made corrections, which we hope to meet with the approval. Considering the comments of Reviewers 2 and 3, we have made substantial changes in the manuscript. Revised portion in the new version were included and are highlighted in yellow in the reviewed manuscript. The following is our point-by-point response to reviewers’ comments:
- General comment: Please make sure to write in italics the Latin names of microorganisms in the whole manuscript and the references.
Suggestion accepted. Thank you
- General comment: The are too many references added in some parts of the manuscript that seem unnecessary. Line 73 has 11 references. Other examples with too many references are lines 53, 240, 252, 257, 279.
Suggestion accepted. Thank you
- Line 32: Correct the capital first letter of “Typhimurium” and use italics. Do the same in lines 75, 94, 115, 37
We had Typhimurium as serovar, but now, we have change it to typhimurium as specie. Thank you
- Lines 74-86: The description how the Authors’ “laboratory” created this necrotic enteritis challenge model should be limited to the absolutely necessary, as this protocol is simply applied in this manuscript that focuses on Lippia origanoides.
Suggestion accepted. Paragraph was deleted.Thank you
- Table 1: Please revise the descriptions of the feed Ingredients (not “Items”). Write the full forms of SBM and DDGS. Delete numerical values i.e. “9-14-18”, “45.16%”, “8.1%”, “60%”, “0.06%”. Delete “0.5lb/ton”. Explain the composition of the various vitamin, mineral and other premixes (Waldroup TM mix? Tyson 2x Broiler Vit? Optiphos2000? Se Premix? Santoquin)
Suggestion accepted. Table 1 has been edited accordingly. Thank you
- Line 117. Place a dot after PC.
Suggestion accepted. Thank you
- Lines 128-130. Give details about the commercial kits (Names & Producer)
Suggestion accepted. Thank you
- Line 132: Text “n=40” refers to 4 chicken per cage? How were these slaughtered? Separately for those in section 2.3?
Yes, text has been modified for better clarification. Thank you
- Line 137: The abbreviations of these bacteria were already introduced previously in the manuscript. There is no need to repeat them here.
Suggestion accepted. Thank you
- Line 142: Correct “prvious”
Suggestion accepted. Thank you
- Lines 148-159: Give a brief description of the production of the “digested diets” and the inoculation procedure. Explain all uncommon abbreviations (THIO, TSB, TSA).
Suggestion accepted. A brief description of the production of the “digested diets” and the inoculation procedure has been added to the text.Thank you
- Lines 166-177: Explain in more details the situation shown in Table 2. What happened before and after each inoculation?
Suggestion accepted. Thank you
However, the “beneficial” effect of essential oil supplementation although statistically significant is actually quite minimal. At day 25, Group 3 had only 36.67 g improved weight compared to positive control, but it had 323, 04 g lower weight that the negative control.
Suggestion accepted, and text has been modified. Thank you
In addition, it would be very helpful to learn about the mortality of the birds. Were there no dead birds at all in groups 2 and 3 after the triple challenge?
Suggestion accepted. Total mortality has been included in Table 3 and in result section. Thank you.
- Table 2: It would be preferable to only write the superscripts for the parameters that differ significantly (P<0.05). Also, the addition of a new column with the P values of the ANOVA would provide additional info for the readers. Moreover, why did you not include the periods of days “8-14” and “15-25” (i.e. the feed change periods)? Or you could take into consideration periods up to & after day 18, since at day 18 the birds were inoculated with E. maxima.
Table 2 has been modified to include Pooled SEM and P-values. In the present study, performance parameters were focus on the period when chickens were challenge with the EM and CP.
Also, check values “4.8 2” and “12.0 5”, because the have a “space” between the last two digits.
Suggestion accepted. Thank you
- Figure 1: The quality of this figure is low. The text and numbers very small and hard to read. The “error lines” cannot be distinguished from the gray color of the bars. Either revise the figure or present these results in a table.
Suggestion accepted. Figures were removed and information is presented now in Table 3. Thank you
- Table 3: Did you take into consideration in your statistical analysis that the “negative control” has zero variation between its replications (±0.00)? Did you check the homogeneity of the data before applying the ANOVA? Probably a non-parametric test (such as Kruskal–Wallis test) should be applied to these kind of data instead of the ANOVA. Alternatively, a test excluding NC and only testing groups 2 (NC) and 3 (Lippia) should be performed.
Suggestion accepted. Statistics section has been modified accordingly. We also conducted, as suggested for you, a One-Way ANOVA comparing groups 2 and 3, excluding group 1, the NC, and the results were the same. Thank you.
Furthermore, this table combines results from two different procedures. The lesion scores are from the in vivo trial, whereas the C. perfringens are from the in vitro test. Perhaps they should be presented separately, with titles that clearly describe this point.
As suggested by you and Reviewer 3, we eliminated the figures and change the information to Table format. Furthermore, a new table was included that contains morbidity and mortality.
- Discussion section: Do not repeat things that were mentioned in the introduction (for example lines 219-221).
Suggestion accepted. Thank you
- Conclusions: Do not use references and comparisons with other works in the conclusions. Summarize and highlight the importance of your results.
Suggestion accepted. Thank you
Reviewer 3 Report
Dear Corresponding Author,
Please, clearly and in details respond to each comment and doubts listed line by line below.
Title
Please, consider changing the phrase "necrotic enteritis challenge" to "Salmonella Typhimurium, Eimeria maxima, and Clostridium perfringens challenge". From the reviewer's point of view, it is more suitable. Please, choose between the "challenge" or "induce the necrotic enteritis//necrotic enteritis model".
Comment 1 - The reviewer has a huge doubt in terms of the potential conflict of interests, on the one hand the authors declare that (L303) "the research was conducted in the absence of any commercial or financial relationships that could be construed as a potential conflict of interest.", on the other hand, "The research was supported in part by Promitec S.A.". There is no doubt that this part should be expanded and explain in details.
L13 - Is it possible to change private mail, i.e., xochitl_h@yahoo.com, into institutional one?
Simple summary
Comment 2 - Please, reduce the content of this section according to the guidance. The maximum number of words is established at 200.
Abstract
Comment 3 - the same as above.
L30 - "Lippia origanoides" italics
L32 - "Salmonella"; "Eimeria maxima"; italics
L32-33 "C. perfringens" italics
Comment 4- please, do not use the shortcuts without the explanation in this section.
Introduction
L94 - "S. Typhimurium, E. maxima, C. perfringens"
Material and Methods
Comment 5 - The composition of essential oils from Lippia origanoides should be shown in detail. Additionally, the Authors did not mentioned how, and when the experimental factor was added to the diet? Did the exp. factor was added from the first day of age? The micro encapsulation proces should be also provided.
Comment 6 - Details in terms of the preparation of the diets should be added, as well as the physical form.
L105 - "The Authors mentioned that "No antibiotics, coccidiostats or enzymes were added to the feed (Table 1).", however, in Table 1 the OptiPhos was emphasized... please, correct.
L109 - "L. origanoides" italics
L117 - "as the PC" missing full stop.
Comment 7 - Why the exp. was conducted till 25 d, while the C. perfringens was provided on d 22 and 23? Why the Authors did not prolong the trial till 32 d? 32/35 d is commonly used exp. period in the literature.
L125 - "CO2" lower index
L142 - "prvious" please, correct
L142 - "35 %" without space
L160 - the "Data and Statistical Analysis" section is presented in a poor way... it should be highly expanded. Did the Authors test the normal distribution and homogeneity of variance? Which test was used in this case? Why the Authors mentioned only analysis of variance, while in terms of in vitro digestion assay only two groups were used. Additionally, please inform about the usage of non-parametric tests which were used.
Comment 8 - there is no information about the birds housing, and rearing conditions... it should be significantly expanded.
Comment 9 - at the beginning of the Material and Methods section please add ethics statement.
Results
Comment 10 - If the Authors highlighted the significance, changes, improvement, increasing, limiting of sth, there is a need to add the exact p-value. Please, double-check the whole section in this case.
L181-182 - "on a necrotic enteritis model 181 challenge model. " please, correct
L190-195 - There is no possible to check this part due to the lack of information about statistical model. This data should be calculated once again.
Table 2 - there is a lack of standard error of the mean (SEM), as well as p-value is missing for each trait. It should be added. At the moment, the table is not informative...
L210 - "a necrotic enteritis model challenge model" please, correct
L210-211 - "A) FITC-d, B) SOD, C) 210 IFN-γ, and D) IgA." please, explain the shortcuts.
Figure 1 - probably the letters was match improperly. Is should be PC - a; NC - b and LO - c. Additionally, the exact p-value should be added. The quality of the figure is not acceptable.
Comment 11 - Sorry for that, but the reviewer was totally misleading, one the L151 the Authors mentioned that in the case of "In-Vitro Digestion Assay" there were two groups... in Table 3 there are 3... It should be clearly presented for future readers. Please, explain.
Table 3 - SEM, as well as p-value should be added.
Discussion
Comment 12 - the discussion section should be expanded and explain each significant parameter measured/determined in the study.
L301-302 - it is probably mistake.
Author Response
Dear Reviewer, #3, thank you very much for the time you have spent on reviewing our manuscript. Your comments are very valuable and helpful for revising our paper and guiding our research. We have studied those comments carefully and have made corrections, which we hope to meet with the approval. Considering the comments of Reviewers 2 and 3, we have made substantial changes in the manuscript. Revised portion in the new version were included and are highlighted in yellow in the reviewed manuscript. The following is our point-by-point response to reviewers’ comments:
Dear Corresponding Author,
Please, clearly and in details respond to each comment and doubts listed line by line below.
Title
Please, consider changing the phrase "necrotic enteritis challenge" to "Salmonella Typhimurium, Eimeria maxima, and Clostridium perfringens challenge". From the reviewer's point of view, it is more suitable. Please, choose between the "challenge" or "induce the necrotic enteritis//necrotic enteritis model".
Suggestion accepted. The title has been modified. Thank you
Comment 1 - The reviewer has a huge doubt in terms of the potential conflict of interests, on the one hand the authors declare that (L303) "the research was conducted in the absence of any commercial or financial relationships that could be construed as a potential conflict of interest.", on the other hand, "The research was supported in part by Promitec S.A.". There is no doubt that this part should be expanded and explain in details.
You are right about this, and as a matter of fact, the Chief Editor asked to all the authors of this work, to sign a Disclosure of Potential Conflicts of Interest Form that has been submitted. Thank you.
L13 - Is it possible to change private mail, i.e., xochitl_h@yahoo.com, into institutional one?
Yes of course. Thank you
Simple summary
Comment 2 - Please, reduce the content of this section according to the guidance. The maximum number of words is established at 200.
Simple summary has 192 words. Thank you.
Abstract
Comment 3 - the same as above.
L30 - "Lippia origanoides" italics
L32 - "Salmonella"; "Eimeria maxima"; italics
L32-33 "C. perfringens" italics
Comment 4- please, do not use the shortcuts without the explanation in this section.
Suggestion accepted. Abstract has been reduced to 199 words. However, due to the maximum number of words established at 200, it is not possible to spell out the full names of the serological biomarkers used. fluorescein isothiocyanate-dextran (FITC-d); superoxide dismutase (SOD); gamma interferon (IFN-γ); Immunoglobulin A (IgA). They would take to much space. Thank you.
Introduction
L94 - "S. Typhimurium, E. maxima, C. perfringens"
Suggestion accepted. Thank you
Material and Methods
Comment 5 - The composition of essential oils from Lippia origanoides should be shown in detail. Additionally, the Authors did not mentioned how, and when the experimental factor was added to the diet? Did the exp. factor was added from the first day of age? The micro encapsulation proces should be also provided.
Suggestion accepted. Information has been incorporated into the text. Thank you
Comment 6 - Details in terms of the preparation of the diets should be added, as well as the physical form.
Suggestion accepted. Information has been incorporated into the text. Thank you
L105 - "The Authors mentioned that "No antibiotics, coccidiostats or enzymes were added to the feed (Table 1).", however, in Table 1 the OptiPhos was emphasized... please, correct.
Suggestion accepted. Thank you.
L109 - "L. origanoides" italics
Suggestion accepted. Thank you.
L117 - "as the PC" missing full stop.
Comment 7 - Why the exp. was conducted till 25 d, while the C. perfringens was provided on d 22 and 23? Why the Authors did not prolong the trial till 32 d? 32/35 d is commonly used exp. period in the literature.
Because this experiment was done in cages. Our battery cages can have up to 10 chickens at 25 days of age. Our IACUC protocol does not allow us to go over this period of time.
L125 - "CO2" lower index
Suggestion accepted. Thank you.
L142 - "prvious" please, correct
Suggestion accepted. Thank you.
L142 - "35 %" without space
Suggestion accepted. Thank you.
L160 - the "Data and Statistical Analysis" section is presented in a poor way... it should be highly expanded. Did the Authors test the normal distribution and homogeneity of variance? Which test was used in this case? Why the Authors mentioned only analysis of variance, while in terms of in vitro digestion assay only two groups were used. Additionally, please inform about the usage of non-parametric tests which were used.
Suggestion accepted. Section has been modified. Thank you.
Comment 8 - there is no information about the birds housing, and rearing conditions... it should be significantly expanded.
Suggestion accepted. Section has been modified. Thank you.
Comment 9 - at the beginning of the Material and Methods section please add ethics statement.
Suggestion accepted. Section has been modified. Thank you.
Results
Comment 10 - If the Authors highlighted the significance, changes, improvement, increasing, limiting of sth, there is a need to add the exact p-value. Please, double-check the whole section in this case.
P value was established with an alpha level of P <0.05. Pool SEM and P values are now shown in Tables 2 and 3. Thank you.
L181-182 - "on a necrotic enteritis model 181 challenge model. " please, correct
Suggestion accepted. Section has been modified. Thank you.
L190-195 - There is no possible to check this part due to the lack of information about statistical model. This data should be calculated once again.
As suggested also for Reviewer 2, statistics section has been modified accordingly. We also conducted, as suggested a One-Way ANOVA comparing groups 2 and 3, excluding group 1, the NC, and the results were the same. Thank you.
Table 2 - there is a lack of standard error of the mean (SEM), as well as p-value is missing for each trait. It should be added. At the moment, the table is not informative...
Suggestion accepted. Tables have been modified. Thank you.
L210 - "a necrotic enteritis model challenge model" please, correct
Suggestion accepted. Figures were change to table form as also suggested by Reviewer 2. Thank you.
L210-211 - "A) FITC-d, B) SOD, C) 210 IFN-γ, and D) IgA." please, explain the shortcuts.
Suggestion accepted. Figures were change to table form as also suggested by Reviewer 2. Moreover, a new table was included to describe morbidity and mortality. Thank you.
Figure 1 - probably the letters was match improperly. Is should be PC - a; NC - b and LO - c. Additionally, the exact p-value should be added. The quality of the figure is not acceptable.
Suggestion accepted. Figures were change to table form as also suggested by Reviewer 2. Thank you.
Comment 11 - Sorry for that, but the reviewer was totally misleading, one the L151 the Authors mentioned that in the case of "In-Vitro Digestion Assay" there were two groups... in Table 3 there are 3... It should be clearly presented for future readers. Please, explain.
On the contrary, we are sorry for the miss leading, the text has been modified. Thank you.
Table 3 - SEM, as well as p-value should be added.
Suggestion accepted. Thank you.
Discussion
Comment 12 - the discussion section should be expanded and explain each significant parameter measured/determined in the study.
Suggestion accepted. Thank you.
L301-302 - it is probably mistake.
Yes. Thank you.
Reviewer 4 Report
The manuscript by Coles et al. represents important results of the research for evaluating the effects of essential oils (Lippia origanoides) as an alternative to antibiotics growth promoters in broiler chickens with NE challenge model. The protocol for NE challenge model is reliable, and materials and methods for data collection are objectives. Results will be of interest to many poultry nutritionists, veterinarians, and producers involved in the poultry industry.
Broad comments
Overall, I think the results and discussions presented in this manuscript are not enough to explain the results of the research. The authors need to present more details of the results on growth performance (3.1) and blood parameters (3.2). Considering the results presented in the manuscript, the results for NC, PC, LEO treatments were not explained well. Authors mentioned the general effects of the essential oils on broiler chickens without or with NE challenge, but they need to discuss the effects of LEO (essential oil) on the growth performance based on the results of other measurements (blood parameters, in vitro digestion assay, NE lesion). Moreover, they need to discuss the mode of actions under the effects of LEO on blood parameters (FITC-dextran, SOD, IgA, IFN-r), in vitro assay, and NE lesion.
There are lots of self-citations for NE challenge model in broiler chickens. Some of the citations are necessary to present the NE challenge model protocol that was published and reliable. However, some of the citations from the same research group (laboratory) are too much throughout the manuscript.
There is a no IACUC approval information in the manuscript. Please provide the IACUC approval information for this research.
Specific comments.
Line 32-33: Italic font style for Salmonella Typhimurium, Eimeria maxima, C. perfringens
Line 37: define the abbreviations (FI, FCR)
Line 55-57: Need references or additional evidence before this sentence (Hence, …….)
Line 64-65: Recommend to use recent data (less than 5 or 10 years) if authors could.
Line 74-91: It may cause a problem of self-citations. The development of NE challenge models is not necessary to include in the introduction section. Authors need to summarize those descriptions and present them in the M&M section.
Line 103-105: Please present which nutrient requirements were referred from NRC (1994) and breeder recommender(Cobb 500).
Line 126: Please provide the temperature for the centrifuge.
Line 127-130. Inappropriate self-citations. Please present the details of the methods for determining FITC-dextran and blood parameters.
Line 151-153: Please define the abbreviations; THIO, TSB
Result (Line:164-200). Please insert P-values when you mentioned the significant effects of the dietary treatments.
Line 166: please use the defined abbreviations: BW, FI, FCR.
Line 170-172 : The BW for LEO treatment had no difference compared to NC treatment on d 7, 14, 18. Also, the BW for PC (d18) was similar to NC and LEO as well. Considering those results, please rewrite the result.
Line 178: Please change the bullet to “Effect of Leo on Intestinal Integrity, Antioxidant, and Anti-inflammatory of Chicken Challenged with NE Organisms”.
Line 186-189: Please add more details about the results (SOD, IFN-r, IgA) for NC, PC, LEO treatments.
Line 194: Please delete double-space and add period “.”.
Table 2. BW(d0), FI (d0-7), FCR (d0-7) had no significant difference among the dietary treatments, but superscripts were presented. It may cause confusion to readers. Please deleted the superscripts for the results without significant difference (P>0.05).
Figure 1-D. Superscript “b” and “c” need to be changed.
Line 219-240: Inappropriate discussion to explain the results of the present study.
Line 245-251: Please do additional discussion for the result of growth performance based on the results of Table 3 and Figure 1
Line 256-259: Please add further discussion on how PC increased the permeability (FITC-dextran), and LEO decreased permeability compared to that of PC.
Line266: It will not be a correct comparison (Probiotics). The reference for probiotics is not suitable to compare the effects of essential oil.
Line 266-267. Could authors mention that secretory IgA is a biomarker to an inflammation under the infection? Staley et al. [65] discussed the sIgA as a biomarker to evaluate stress and animal welfare. I think there is a gap between animals under the infection and animals for animal welfare research. Please cite a peer-reviewed paper that is more suitable for your experimental conditions.
Line 276-288: Please rewrite the conclusion section. Please remove the references in the conclusion section and present the conclusion based on the results in the present study. Some limitations (Line 283-288) of this study should be included in the discussion section.
References: Match the same way to present “doi information” (ex. doi:10.0000/00000).
Table 1: Please provide the spec or information of “Waldroup TM mix, Tyson 2x Broiler Vit, OptiPhos2000, Se Premix, Santoquin” under the table. If authors could, please provide the analyzed value of the diets as well.
Table 2. The font style of “P” value is different compared to that for Figure 1. Please check again and use the same “P” throughout the manuscript based on the journal’s guideline.
Author Response
Dear Reviewer, #4, thank you very much for the time you have spent on reviewing our manuscript. Your comments are very valuable and helpful for revising our paper and guiding our research. We have studied those comments carefully and have made corrections, which we hope to meet with the approval. Considering the comments of Reviewers 2 and 3, we have made substantial changes in the manuscript. Revised portion in the new version were included and are highlighted in yellow in the reviewed manuscript. The following is our point-by-point response to reviewers’ comments:
Broad comments
Overall, I think the results and discussions presented in this manuscript are not enough to explain the results of the research. The authors need to present more details of the results on growth performance (3.1) and blood parameters (3.2). Considering the results presented in the manuscript, the results for NC, PC, LEO treatments were not explained well. Authors mentioned the general effects of the essential oils on broiler chickens without or with NE challenge, but they need to discuss the effects of LEO (essential oil) on the growth performance based on the results of other measurements (blood parameters, in vitro digestion assay, NE lesion). Moreover, they need to discuss the mode of actions under the effects of LEO on blood parameters (FITC-dextran, SOD, IgA, IFN-r), in vitro assay, and NE lesion.
There are lots of self-citations for NE challenge model in broiler chickens. Some of the citations are necessary to present the NE challenge model protocol that was published and reliable. However, some of the citations from the same research group (laboratory) are too much throughout the manuscript.
There is a no IACUC approval information in the manuscript. Please provide the IACUC approval information for this research.
Suggestions accepted. Thank you.
Specific comments.
Line 32-33: Italic font style for Salmonella Typhimurium, Eimeria maxima, C. perfringens
We had Typhimurium as serovar, but now, we have change it to typhimurium as specie. Thank you
Line 37: define the abbreviations (FI, FCR)
Suggestions accepted. Thank you.
Line 55-57: Need references or additional evidence before this sentence (Hence, …….)
Suggestions accepted. Thank you.
Line 64-65: Recommend to use recent data (less than 5 or 10 years) if authors could.
Suggestions accepted. Thank you.
Line 74-91: It may cause a problem of self-citations. The development of NE challenge models is not necessary to include in the introduction section. Authors need to summarize those descriptions and present them in the M&M section.
Suggestions accepted. Thank you.
Line 103-105: Please present which nutrient requirements were referred from NRC (1994) and breeder recommender(Cobb 500).
Suggestions accepted. Table 1 has been modified. Thank you.
Line 126: Please provide the temperature for the centrifuge.
Suggestions accepted. Thank you
Line 127-130. Inappropriate self-citations. Please present the details of the methods for determining FITC-dextran and blood parameters.
Suggestions accepted. Thank you
Line 151-153: Please define the abbreviations; THIO, TSB
Suggestions accepted. Thank you
Result (Line:164-200). Please insert P-values when you mentioned the significant effects of the dietary treatments.
Suggestions accepted. Thank you
Line 166: please use the defined abbreviations: BW, FI, FCR.
Suggestions accepted. Thank you
Line 170-172 : The BW for LEO treatment had no difference compared to NC treatment on d 7, 14, 18. Also, the BW for PC (d18) was similar to NC and LEO as well. Considering those results, please rewrite the result.
Suggestions accepted. Thank you
Line 178: Please change the bullet to “Effect of Leo on Intestinal Integrity, Antioxidant, and Anti-inflammatory of Chicken Challenged with NE Organisms”.
Suggestions accepted. Thank you
Line 186-189: Please add more details about the results (SOD, IFN-r, IgA) for NC, PC, LEO treatments.
Suggestions accepted. Thank you
Line 194: Please delete double-space and add period “.”.
Suggestions accepted. Thank you
Table 2. BW(d0), FI (d0-7), FCR (d0-7) had no significant difference among the dietary treatments, but superscripts were presented. It may cause confusion to readers. Please deleted the superscripts for the results without significant difference (P>0.05).
Suggestions accepted. Thank you
Figure 1-D. Superscript “b” and “c” need to be changed.
Figures have been eliminated, data is presented in Table format now. Thank you
Line 219-240: Inappropriate discussion to explain the results of the present study.
Suggestions accepted. Thank you
Line 245-251: Please do additional discussion for the result of growth performance based on the results of Table 3 and Figure 1
Suggestions accepted. Thank you
Line 256-259: Please add further discussion on how PC increased the permeability (FITC-dextran), and LEO decreased permeability compared to that of PC.
Suggestions accepted. Thank you
Line266: It will not be a correct comparison (Probiotics). The reference for probiotics is not suitable to compare the effects of essential oil.
Suggestions accepted. Thank you
Line 266-267. Could authors mention that secretory IgA is a biomarker to an inflammation under the infection? Staley et al. [65] discussed the sIgA as a biomarker to evaluate stress and animal welfare. I think there is a gap between animals under the infection and animals for animal welfare research. Please cite a peer-reviewed paper that is more suitable for your experimental conditions.
Suggestions accepted. Thank you
Line 276-288: Please rewrite the conclusion section. Please remove the references in the conclusion section and present the conclusion based on the results in the present study. Some limitations (Line 283-288) of this study should be included in the discussion section.
Suggestions accepted. Thank you
References: Match the same way to present “doi information” (ex. doi:10.0000/00000).
Suggestions accepted. Thank you
Table 1: Please provide the spec or information of “Waldroup TM mix, Tyson 2x Broiler Vit, OptiPhos2000, Se Premix, Santoquin” under the table. If authors could, please provide the analyzed value of the diets as well.
Suggestions accepted. Thank you
Table 2. The font style of “P” value is different compared to that for Figure 1. Please check again and use the same “P” throughout the manuscript based on the journal’s guideline.
Suggestions accepted. Thank you
Round 2
Reviewer 2 Report
- Lines 227-233 and Table 3: It seems that there was a very high mortality before day 18 (NC 14%, PC 20%, Lip.Orig. 19%). What is the reason for the high mortality? What about the mortality of the negative control group? What was the age of the dead birds? How were you sure that is was not “… due to NE”?
- Table 3: Please correct “Mrobidity”.
Author Response
Reviewer 2
Comments and Suggestions for Authors
- Lines 227-233 and Table 3: It seems that there was a very high mortality before day 18 (NC 14%, PC 20%, Lip.Orig. 19%). What is the reason for the high mortality? What about the mortality of the negative control group? What was the age of the dead birds? How were you sure that is was not “… due to NE”?
Yes, for this study, as you can see, all three groups arrived at day 18 with an average of 80 chickens, when we started the trial with 100. The reason for that high mortality was that we received chickens with severe yolk sac infection and omphalitis, due to a poor quality hatching. Chicken that did not died during the first seven days, had to be removed. Hence, the most important information for the present study, was the chickens that died following the challenge of NE and CP and the cause of death was due to necrotic enteritis.
- Table 3: Please correct “Mrobidity”.
Suggestion accepted. Thank you.

Reviewer 3 Report
Dear Corresponding Author,
The reviewer wants to thank you for including the suggested corrections.
Please, find the additional comments and suggestions listed line by line below.
L54 - "C. perfringens", is mentioned for the first time so the full Latin name should be written with the shortcut, i.e., (CP).
L57 - "Clostridium perfringens", here it should be used a short name
L66 - please, reconsider "Eimeria spp. infections" instead of "coccidia infections"
Table 1 - please, add the soybean meal crude protein level; additionally, reverse the upper indexes in mineral premix (a) and vitamin premix (b); please, correct the upper indexes in the footer.
Comment 1 - Still in the material and methods section there is no detailed information about the experimental factor. Please, add the essential oil composition. The addition of activity of the experimental factor will be highly appreciated.
L134 - ill instead of "Ileal"
Table 2 - The Authors should double-check the calculations because the FCR values are not correct. Thus, the statistical analyses could contain the error. Please, double-check the other parameters.
L301 - "showed a significant improvement of BW" from the reviewer's point of view it is not true. In the current study, no growth performance results were improved. The reduction of the harmful effects of induced infection/dysbiosis was observed.
Comment 2 - the Reviewer is impressed in the case of the honesty of the Authors but does the highlight of the first sentence in the conclusions is necessary? Or is it possible to remove it?
Author Response
Reviewer 3
Dear Corresponding Author,
The reviewer wants to thank you for including the suggested corrections.
Please, find the additional comments and suggestions listed line by line below.
L54 - "C. perfringens", is mentioned for the first time so the full Latin name should be written with the shortcut, i.e., (CP).
Suggestion accepted. Thank you.
L57 - "Clostridium perfringens", here it should be used a short name
Suggestion accepted. Thank you.
L66 - please, reconsider "Eimeria spp. infections" instead of "coccidia infections"
Suggestion accepted. Thank you.
Table 1 - please, add the soybean meal crude protein level; additionally, reverse the upper indexes in mineral premix (a) and vitamin premix (b); please, correct the upper indexes in the footer.
Suggestion accepted. Thank you.
Comment 1 - Still in the material and methods section there is no detailed information about the experimental factor. Please, add the essential oil composition. The addition of activity of the experimental factor will be highly appreciated.
Suggestion accepted. The qualitative and quantitative chemical composition of Lippia origanoides essential oils of LEO is now summarized in Table 1. The LEO sample was submitted to chromatographic analysis 7890A (Laboratory of chromatography and mass spectrometry Industrial University of Santander, Bucaramanga, Colombia). The sample showed 16 compounds, however, Carvacol and thymol were the compounds with the highest content (Table 1). Thank you.
L134 - ill instead of "Ileal"
Intestinal lesions of NE are mainly focus in the ileum. However we have change the word “Ileal” for intestinal NE lesions. Thank you.
Table 2 - The Authors should double-check the calculations because the FCR values are not correct. Thus, the statistical analyses could contain the error. Please, double-check the other parameters.
Thank you so much for your observation. We did had two miscalculations for FCR on d 0-7 and d 8-25, we have made the corrections. However, even with the modifications the tendency of the results did not change the end point of the results. Thank you.
L301 - "showed a significant improvement of BW" from the reviewer's point of view it is not true. In the current study, no growth performance results were improved. The reduction of the harmful effects of induced infection/dysbiosis was observed.
You are right about that, and we concurred with your observation. The text has been modified accordingly. Thank you.
Comment 2 - the Reviewer is impressed in the case of the honesty of the Authors but does the highlight of the first sentence in the conclusions is necessary? Or is it possible to remove it?
We appreciate indeed you comment, and based on your previous suggestions, we have modified the conclusions accordingly. Thank you very much.

Reviewer 4 Report
Dear authors,
Thanks for the author's revision.
Since the authors did not mention the new line of their revision, it was hard to find the author's response in the manuscript for the suggestions and comments by the reviewer. And authors respond to all of the reviewer's comments (Suggestions accepted. Thank you), but some of the comments or suggestions have not been made.
I would appreciate it if you let me know how my comments were addressed in the next revision.
(1) Please define the abbreviations in the abstract section (ex. PC, BW, BWG, FITC-d, SOD, IFN-r, IgA)
(2)Line 127: Please provide the temperature for the centrifuge.
(3) Result (Line:164-200). Please insert P-values when you mentioned the significant effects of the dietary treatments.
(4) Table 4. The superscripts “b” and “c” should be changed each other for IgA result.
(5) Line 245-251: Please do additional discussion for the result of growth performance based on the results of Table 3, table 4.
(6) Line266: It will not be a correct comparison (Probiotics). The reference for probiotics is not suitable to compare the effects of essential oil.
(7) Line 266-267. Could authors mention that secretory IgA is a biomarker to an inflammation under the infection? Staley et al. [65] discussed the sIgA as a biomarker to evaluate stress and animal welfare. I think there is a gap between animals under the infection and animals for animal welfare research. Please cite a peer-reviewed paper that is more suitable for your experimental conditions.
References: Match the same way to present “doi information” (ex. doi:10.0000/00000).
- Adhikari, P.; Kiess, A.; Adhikari, R.; Jha, R. An approach to alternative strategies to control avian 395 coccidiosis and necrotic enteritis. J. Appl. Poult. Res. 2020, 29, 515–534. https://doi.org/10.1016 396 /j.japr.2019.11.005.
Table 1: If authors could, please provide the analyzed value of the diets as well.
Author Response
Reviewer 4
Dear authors,
Thanks for the author's revision.
Since the authors did not mention the new line of their revision, it was hard to find the author's response in the manuscript for the suggestions and comments by the reviewer. And authors respond to all of the reviewer's comments (Suggestions accepted. Thank you), but some of the comments or suggestions have not been made.
I would appreciate it if you let me know how my comments were addressed in the next revision.
Of course, and we are sorry for not being clear enough. The new version contains the corrections highlighted in blue for better clarification. Thank you very much.
(1) Please define the abbreviations in the abstract section (ex. PC, BW, BWG, FITC-d, SOD, IFN-r, IgA)
Suggestion accepted. Abstract has been modified. Thank you.
(2)Line 127: Please provide the temperature for the centrifuge.
Suggestion accepted. The temperature has been added into the text. Thank you.
(3) Result (Line:164-200). Please insert P-values when you mentioned the significant effects of the dietary treatments.
For the statistical analysis, performance variables, each group had ten replicates and ten chickens in each replicate (n=10 replicates/group; 10 chickens/replicate). BW and BWG were performed in individual chickens/group. FI and FCR were calculated by cage bases (n=10). The number of samples per variable evaluated was n= 5 for in vitro proliferation of CP; n= 40 for NE lesion score; n= 20 for serum variables. Hence, all these variables implied a normal distribution (Shapiro-Wilk test), and the homoscedasticity was verified (Levene's test). Accordingly, the parametric test of analysis of variance as a completely randomized design, using the General Linear Models procedure of SAS was used [36]. Significant differences among the means were determined by Duncan's multiple range test at P < 0.05, and the P value was established with an alpha level of P < 0.05. Total mortality were compared by a chi-square test of independence, testing all possible combinations to determine the significance with an alpha level of P < 0.05.
P-values are included in all tables. Thank you.
(4) Table 4. The superscripts “b” and “c” should be changed each other for IgA result.
Suggestion accepted. Superscripts have been changed. Thank you.
(5) Line 245-251: Please do additional discussion for the result of growth performance based on the results of Table 3, table 4.
We have added a sentence and references, but we do not feel that the result section should be use to discuss results. Instead, we have modified the discussion section to cover your suggestion in the reviewed file (Lines 310-314). Thank you.
(6) Line266: It will not be a correct comparison (Probiotics). The reference for probiotics is not suitable to compare the effects of essential oil.
Suggestion accepted. Reference and comparison were removed. Thank you.
(7) Line 266-267. Could authors mention that secretory IgA is a biomarker to an inflammation under the infection? Staley et al. [65] discussed the sIgA as a biomarker to evaluate stress and animal welfare. I think there is a gap between animals under the infection and animals for animal welfare research. Please cite a peer-reviewed paper that is more suitable for your experimental conditions.
Suggestion accepted. Reference more suitable for that statement have been added. Thank you.
References: Match the same way to present “doi information” (ex. doi:10.0000/00000).
- Adhikari, P.; Kiess, A.; Adhikari, R.; Jha, R. An approach to alternative strategies to control avian 395 coccidiosis and necrotic enteritis. J. Appl. Poult. Res. 2020, 29, 515–534. https://doi.org/10.1016 396 /j.japr.2019.11.005.
Suggestion accepted. References have been corrected. Thank you.
Table 1: If authors could, please provide the analyzed value of the diets as well.
Calculated analysis of the diet is presented in Table 1. Thank you.
